# A Predictive Pharmacokinetic Model for Immune Cell-Mediated Uptake and Retention of Nanoparticles in Tumors

**DOI:** 10.3390/ijms232415664

**Published:** 2022-12-10

**Authors:** Ailton Sousa-Junior, Chun-Ting Yang, Preethi Korangath, Robert Ivkov, Andris Bakuzis

**Affiliations:** 1Instituto de Física, Universidade Federal de Goiás, Goiânia 74690-900, GO, Brazil; 2FarmaTec—Laboratório de Tecnologia Farmacêutica, Universidade Federal de Goiás, Goiânia 74690-631, GO, Brazil; 3Department of Radiation Oncology and Molecular Radiation Sciences, Johns Hopkins University School of Medicine, Baltimore, MD 21218, USA; 4Department of Oncology, Johns Hopkins University School of Medicine, Baltimore, MD 21218, USA; 5Department of Mechanical Engineering, Whiting School of Engineering, Johns Hopkins University, Baltimore, MD 21218, USA; 6Department of Materials Science and Engineering, Whiting School of Engineering, Johns Hopkins University, Baltimore, MD 21218, USA; 7CNanoMed, Universidade Federal de Goiás, Goiânia 74690-631, GO, Brazil

**Keywords:** cancer nanomedicine, targeted delivery, immune cell interactions, macrophages, iron oxide nanoparticles, tumor targeting, tumor microenvironment

## Abstract

A promise of cancer nanomedicine is the “targeted” delivery of therapeutic agents to tumors by the rational design of nanostructured materials. During the past several decades, a realization that in vitro and in vivo preclinical data are unreliable predictors of successful clinical translation has motivated a reexamination of this approach. Mathematical models of drug pharmacokinetics (PK) and biodistribution (BD) are essential tools for small-molecule drugs development. A key assumption underlying these models is that drug-target binding kinetics dominate blood clearance, hence recognition by host innate immune cells is not explicitly included. Nanoparticles circulating in the blood are conspicuous to phagocytes, and inevitable interactions typically trigger active biological responses to sequester and remove them from circulation. Our recent findings suggest that, instead of referring to nanoparticles as designed for active or passive “tumor targeting”, we ought rather to refer to immune cells residing in the tumor microenvironment (TME) as active or passive actors in an essentially “cell-mediated tumor retention” process that competes with active removal by other phagocytes. Indeed, following intravenous injection, nanoparticles induce changes in the immune compartment of the TME because of nanoparticle uptake, irrespective of the nature of tumor targeting moieties. In this study, we propose a 6-compartment PK model as an initial mathematical framework for modeling this tumor-associated immune cell-mediated retention. Published in vivo PK and BD results obtained with bionized nanoferrite^®^ (BNF^®^) nanoparticles were combined with results from in vitro internalization experiments with murine macrophages to guide simulations. As a preliminary approximation, we assumed that tumor-associated macrophages (TAMs) are solely responsible for active retention in the TME. We model the TAM approximation by relating in vitro macrophage uptake to an effective macrophage avidity term for the BNF^®^ nanoparticles under consideration.

## 1. Introduction

Cancer nanomedicine exists at the intersection of medicine, biology, and materials science. A considerable effort has been devoted to developing “targeted” nanoparticles that, after intravenous administration, accumulate preferentially and selectively on cancer cell membranes or in tumor interstitial spaces to deliver a cytotoxic payload. In general, strategies to target nanoparticles are either “passive” or “active”, with the former aiming to tailor nanoparticle physicochemical properties to enhance their retention in the tumor microenvironment (TME) by exploiting aberrant tumor physiology. On the other hand, active targeting aims to increase retention in the TME by chemical modification of the nanoparticle surface to exploit high binding affinity with molecular target(s) uniquely or highly expressed on cancer cell membranes or in the TME [1,2,3]. By some accounts, expectations for nanoparticle targeting have not materialized, prompting a reevaluation of discrepancies between preclinical predictions and clinical performance [4]. One barrier to progress has been a reliance on paradigms and models originally developed for small-molecule drug delivery.

Compartment models are standard models used to describe the pharmacokinetics (PK) of small molecule drugs [5]. These are often modified to study the PK of various nanoparticle formulations such as solid lipid nanoparticles or lipid coated nanoparticles for drug delivery [6,7], sometimes with imaging [8], and metallic nanoparticles [9]. Typically, these model the body of the host as a set of interconnected compartments upon which first-order kinetics are imposed to predict the rate at which mass transfer occurs between compartments. If the drug rapidly distributes throughout the body after intravenous (i.v.) injection, the model assumes that the entire organism is a single kinetically homogeneous compartment. However, considering their blood concentrations over time, most drugs exhibit two distinct PK phases: in the first, distributive phase, the blood concentration of the drug decreases rapidly; and in the second, post-distributive phase, the blood concentration of the drug decreases more slowly. For many small-molecule drugs, at least two compartments are required to appropriately model their PK properties: a central compartment, representing blood and highly perfused organs, and a peripheral compartment, representing all other tissues and organs [10].

For nanoparticles intended for treating solid tumors, the standard model was adapted to include a third compartment to represent nanoparticle PK, and uptake by a tumor [11]. Specifically, adding a tumor compartment to the central and peripheral compartments distinguishes tumor specific uptake from nonspecific uptake by healthy peripheral tissues and organs. Using this adaptation, Wong et al. quantitatively compared the PK of doxorubicin with that of Doxil^®^, a liposomal formulation of doxorubicin, the first FDA-approved nanodrug [11,12]. Later, Thurber and Wittrup modified one of their earlier models to describe the systemic delivery of antibodies to a tumor, obtaining a straightforward mechanistic compartment model [13], which was also used to predict the tumor uptake of nanoparticles [14]. Sousa-Junior et al. applied this latter adaptation to successfully predict delivery efficiency of erythrocyte membrane-camouflaged magneto-fluorescent nanocarriers in two different mouse tumor models (Ehrlich and Sarcoma 180), before and after the TME was manipulated to enhance tumor uptake [15,16].

Recent reports suggest that either active or passive targeting approaches may be suspect, because active biological processes within stromal compartments of the TME may dominate nanoparticle retention [2,3]. This raises questions about the utility of current approaches used to target cancer cells and casts doubt on the predictions of nanoparticle PK based on standard compartment models. Unlike their small-molecule counterparts, nanoparticles possess physicochemical features, e.g., size, surface charge, ligand chemical groups, etc. that make them conspicuous to circulating and tissue-resident innate immune cells. Korangath et al. demonstrated that antibody-labeled nanoparticles were retained by the immune compartment of the TME, irrespective of the presence of the target receptor on cancer cell membranes [17]. Later, Kingston et al. demonstrated that a subset of tumor endothelial cells facilitates nanoparticle transport into solid tumors [18]. Collectively, a growing body of evidence prompts modifications to standard compartment models in order to account for complex interactions between nanoparticles and living systems [3,19].

The lesson learned from recent studies is that the biology of the host is just as important as the physicochemical properties of nanoparticles when considering nanoparticle PK and biodistribution (BD). If host immune biology determines nanoparticle performance, then the mathematical models must evolve accordingly. Here, we extend the 3-compartment model by subdividing each of the three major compartments into two sub-compartments. We consider one sub-compartment to describe “noninternalized” nanoparticles, i.e., nanoparticles trapped within the extracellular environment of (1) blood and highly perfused tissues, (2) healthy peripheral tissues, and (3) the TME. The second sub-compartment describes “internalized” nanoparticles, i.e., nanoparticles internalized within each of these three major compartments by a specific set of innate immune cells (e.g., macrophages).

The concept of dividing each major compartment into the two sub-compartments was motivated by demonstrations that nanoparticles labeled with either cancer-specific antibodies (designed for active targeting) or unlabeled (designed for passive targeting) were similarly retained among different cancer tumor models, irrespective of the presence of molecular targets on the cells, or the nature of the targeting moiety on the nanoparticles [2]. Instead, the mechanism(s) dominating the retention of both types of nanoparticles was determined to be their internalization by tumor-associated innate immune cells. Those results provide evidence that concepts of nanoparticle targeting may require revision. It also suggests that host cells other than the intended target cells can engage with the nanoparticles to affect nanoparticle fate [17,18,19].

In this study, we assume that macrophages are solely responsible for the cell-mediated tumor retention of nanoparticles as a first approximation to reflect the widely accepted role of circulating and resident macrophages in nanoparticle clearance. Notwithstanding the obvious limitations of this assumption, our primary goal with this new six-compartment PK model is to begin to develop a new mathematical framework that explicitly accounts for biological mechanisms that recognize and interact with nanoparticles in a way that affects retention within the TME. We combined data obtained from in vitro studies of macrophage internalization of bionized nanoferrite^®^ (BNF^®^) nanoparticles with published in vivo PK and BD results [20], to derive intercompartment exchange rate constants $K_{ij}$ (with *i* and *j* being *b* for blood, *p* for peripheral, or *t* for tumor), and other PK modeling parameters to describe in vivo nanoparticle fate following systemic delivery.

## 2. Results

### 2.1. $K_{pb}$, $K_{el}$ and $K_{bp}$ from In Vivo PK and BD Data

Model and its development, and fitting strategy are described in Materials and Methods. The fitted values of $\lambda_{b}$, $\lambda_{p}$, $K_{pb}$, $K_{el}$ and $K_{bp}$ obtained are summarized in Table 1, and the results of fitting to PK data are shown in Figure 1a. Since the nanoparticles used in the present study have a mean core diameter of about 100 nm (Table 1), the corresponding values of $K_{pb}$, $K_{el}$ and $K_{bp}$ (Table 1) were adopted for all simulations hereafter. BD data are adapted with permission here from Natarajan et al. [20] (Copyright 2008 American Chemical Society) for comparison and convenience.

### 2.2. $K_{bt}$ and $K_{tb}$ from In Vivo BD Data 

The addition of the tumor compartment requires the addition of two new rate constants to the model, namely $K_{bt}$ and $K_{tb}$ which must be estimated from BD data. The BD data used for this are reproduced from Ref. [20] in Figure 1b, where both tumor and peripheral tissues should be considered. Values for these rate constants were 0.0011 and 0.0088 h^−1^, and were chosen to match simulations with reported data. We note that the BD data, originally reported in percent injected dose (%ID)/g, were converted to %ID by taking a mean tumor volume of 225 mm^3^ as originally reported, and by assuming a tumor density of approximately 1 g/cm^3^. 

### 2.3. $K_{im}$ and $K_{mi}$ from In Vitro Data 

The fitting of the in vitro nanoparticle uptake data in macrophages was performed considering Equation (17) (Materials and Methods) using either mean values or fitting to all values. The results showed that differences between the two analyses were within variance (Table 2, Table 3 and Table 4 and Appendix A). A full analysis is shown graphically in Appendix A. To conserve space, we show the results of the analysis from mean values in Figure 2. Generally, M1 macrophages displayed the highest internalization kinetics, $K_{in}$, and the lowest expulsion, $K_{out}$, for each of the three BNF^®^ nanoparticle configurations tested. Indeed, $K_{out}$ = 0 for M1 except BNF-IgG, which was still lower than for either M0 or M2 (Figure 2; Table 2, Table 3 and Table 4). For simulations involving BNF-Plain, the values chosen for $K_{bm}$ and $K_{mb}$ were those found for $K_{in}$ and $K_{out}$ for the M0 polarization state, 0.00092 ± 0.00051 and 0.11562 ± 0.07059, respectively, whereas the values chosen for $K_{pm}$ and $K_{mp}$, as well as for $K_{tm}$ and $K_{mt}$ were those found for $K_{in}$ and $K_{out}$ at the M1 polarization state, 0.00101 ± 0.00015 and 0.00000 ± 0.01380, respectively. 

### 2.4. Results for $k_{b}$, $k_{p}$, and $k_{t}$ from Simulations 

Simulation results obtained for 100-nm BNF-Plain nanoparticles with the foregoing values determined for $K_{pb}$, $K_{el}$, $K_{bp}$, $K_{bt}$, $K_{tb}$, $K_{bm}$, $K_{mb}$, $K_{pm}$, $K_{mp}$, $K_{tm}$, and $K_{mt}$, with the multiplication factors $k_{b}$, $k_{p}$, and $k_{t}$ all set to 1, are deceptive because they suggest that macrophages play a negligible role in the tumor retention of nanoparticles, which contradicts published reports. In contrast, by setting $k_{b}$, $k_{p}$, and $k_{t}$ to 40, we were able to model the dominant role of immune cells (represented solely by macrophages in this study, as a first approximation) in the tumor retention of 100-nm BNF-Plain nanoparticles. The resulting simulations for both tumor and peripheral retentions of 100-nm BNF-Plain, BNF-Her, and BNF-IgG nanoparticles are displayed in Figure 3, with dashed lines representing passive retention, solid-colored lines representing active retention by macrophages, and solid-gray lines representing the sum of passive and active retentions.

### 2.5. Effect of Parameters on the Immune Cell-Mediated Compartment Model 

Fixing most of the parameters of the BNF-Her nanoparticle, simulated PK intratumor, blood and peripheral delivery efficiencies were evaluated by varying model parameters to investigate the role of passive and active delivery (Figure 4). The effect of blood PK was evaluated by increasing $K_{pb}$ by a factor A, while in decreasing $K_{pm}$ we investigated the importance of the peripheral-macrophage rate constant. The role of macrophage M1 polarization, in particular the exocytosis of nanoparticles, was evaluated by varying $K_{out}$. Finally, comparisons with a passive model (Figure 5b) are made by setting $k_{t}=k_{b}=k_{p}=0$, i.e., no active uptake by immune cells.

## 3. Discussion

Mathematical models provide predictive capabilities for rational product design and descriptive insights to illuminate underlying mechanisms. Predictive accuracy requires knowledge of underlying processes to be faithfully incorporated into the models. However, even incomplete models can provide insights into our understanding of underlying mechanisms and their interactions by comparing experimental results with model predictions. In this regard, mathematical models are indispensable tools to aid in the understanding of complex phenomena, such as nanoparticle interactions with living systems. Strategies designed to engineer nanoparticles that exploit active and/or passive targeting to solid tumors rely on an incomplete paradigm of tumor biology, i.e., enhanced permeability and retention (EPR). EPR does not explicitly account for host immune biology or active cell participation in tumor retention.

In the present study, we provide a mathematical framework to model tumor-associated immune cell-mediated retention mechanism(s), with a focus on macrophages. We modified the three-compartment model [15] by further subdividing each compartment to model passive and cell-mediated active retention of nanoparticles within tumor and peripheral tissues. Cell-mediated active retention emphasizes the role of tissue-associated phagocytic immune cells. The final six-compartment PK model is depicted in Figure 5c.

The input values we used for numeric simulations were the time rate constants *K*, and dimensionless multiplication factors *k*, which appear as constant coefficients in the system of ODEs given by Equations (1)–(6). We estimated values governing clearance to peripheral tissue through a 2-compartment PK analysis of published data relevant to the BNF^®^ nanoparticles used in this study. This limited use of published data was necessary because nanoparticle physicochemical properties (i.e., composition, size, charge, shape, etc.) affect their PK and BD, making each nanoparticle unique. In other words, PK and BD modeling of a specific nanoparticle construct requires in vitro and in vivo data specific for that nanoparticle. It is, therefore, an underlying assumption in our scheme that in vitro nanoparticle uptake data need to be collected for each nanoparticle type and cell lineage to obtain the relevant information for accurate modeling. We note that the size, charge, and other gross physicochemical features of the nanoparticles used in the present study are similar to those used by Natarajan et al., yet the nanoparticles were not identical. Thus, we might attribute some variance of model predictions to these differences. With these assumptions and constraints, and assuming that both tumor and peripheral retentions are governed by similar active or passive mechanisms, we assigned values of *K* for tumor as a relative percentage of those for peripheral tissues with the actual value depending on the specific tumor model guided by pilot BD data (Figure 1b).

Using the aforementioned assumption, we hypothesized that the in vitro internalization kinetics of nanoparticles by immune cells provides reasonable values of *K* describing in vivo uptake by the cell-associated sub-compartment. As a first approximation, we conducted a series of in vitro experiments using only macrophages. Interestingly, we found that independent of the nanoparticle coating, M1 macrophages ingest more nanoparticles than those displaying either the M0 or M2 phenotypes (Figure 2 and Appendix A). Under the conditions tested, and for the RAW264.7 cells, the presence of a (humanized) antibody on the nanoparticle influences uptake, with a higher uptake found for BNF-Her (monoclonal) followed by BNF-IgG (polyclonal) and BNF-Plain (no antibody). We note that a similar phenomenological equation has been suggested to investigate uptake of nanoparticles by macrophages [21]. 

We then used these values to model the behavior of circulating (M0), peripheral resident (M2), and tumor-associated macrophages (M1/M2) (Equation (17)). Assuming that the M0 phenotype (i.e., monocytes) prevails in blood, whereas the M1 phenotype can be in both peripheral and tumor tissues, the values of $K_{bm}$, $K_{mb}$, $K_{pm}$, $K_{mp}$, $K_{tm}$, and $K_{mt}$ were chosen from Table 2. One limitation of this approach is that murine macrophages possess Fc receptors that react to the human Fc region of humanized antibodies, and that the disease state, i.e., inflammation or injury, can skew the M1/M2 balance. Furthermore, tumors can display a range of states between immunologically “hot” (e.g., more M1) or “cold” (e.g., more M2) which varies by type of tumor, stage and size, host immune status, etc. In other words, tumors display a range of states along a continuum that is a dynamic flux between immune suppression and activation, highlighting the need to use caution when interpreting data from mathematical models that are based on limited data from a specific tumor model. 

A first attempt to model tumor-associated immune cell-mediated retention without the multiplication factors $k_{b}$, $k_{p}$, and $k_{t}$ led to results contradicting in vivo data. This suggests that our initial model failed to account for all possible active cell-based retention. By varying $\;k_{b}$, $k_{p}$, and $k_{t}$ we reveal potential effects of the modeled active macrophage uptake for BNF-Plain and BNF-Her nanoparticles (Figure 3). In contrast, according to the simulations, a passive mechanism seems to dominate BNF-IgG nanoparticle accumulation.

The discrepancy between initial model results with the experiment was corrected by the inclusion of a multiplication factor, indicating that in vitro kinetics data obtained from a single cell lineage and fixed numbers (~1 × 10^6^) may be inadequate. While this highlights a potential limitation of the current study, another interpretation may be that in vivo kinetics are faster than in vitro, or that numbers and types of immune cells involved in vivo is time-, tissue-, and/or dose-dependent. Moreover, the need for these multiplication factors can simply be an indication that the macrophage-approximation must be refined further with data collected from in vivo experiments. The potential effect of these possibilities bears further investigation.

In Figure 4 we show results from simulations extending our evaluation of the relative influence of various model parameters on uptake using the BNF-Her NPs as a test case. Here several parameters obtained previously for BNF-Her were held constant, while others were varied. For instance, increasing $K_{pb}$, which prolongs blood circulation time, resulted in an expected higher intratumor delivery (see Figure 4a). Naturally, the PK profile affects tumor uptake mainly by affecting peripheral retention with higher NP uptake in the peripheral compartment (rapid clearance) corresponding to a lower tumor uptake. This can be interpreted as less interaction with or “visibility” to the circulating macrophage population. However, when we included the in vitro kinetic data, the opposite was true, indicating substantial interactions. While this was expected, the model prediction based on those results overestimates clearance. 

Similar behavior occurs with increasing both $K_{bt}$ and $K_{tb}$, which were assumed to be a factor of $K_{bp}$ and $K_{pb}$, respectively (Appendix A). Recently, it was suggested that specific tumor endothelial cells are involved in nanoparticle transport into solid tumors [18]. Since the endocytosis of nanoparticles depends on several parameters such as membrane receptors, nanoparticle size, shape, surface charge and coating layer [22,23], an increased intra-tumor distribution (ITD) for BNF-Her could be related to improved transport into the TME due to coating. 

Some parameters affect the passive versus active delivery mechanism, where a strong effect was found when $K_{out}$ of macrophages in the M1 state was varied. Enhancing exocytosis of the nanoparticles decreased the importance of active retention in comparison to passive retention (Figure 4b), as found for BNF-IgG NPs. Furthermore, in certain cases, the model predicts a peak for ITD at a short time from passive accumulation. This has not been reported, perhaps because most data are obtained after sacrifice and are thus snapshots of the kinetics at specific time points. Non-invasive techniques such as magnetic particle imaging or alternating current biosusceptometry might show this model prediction [24,25].

Decreasing the capacity of immune cells to internalize nanoparticles also affects the BD/PK. Curiously, the model predicts that in this case tumor uptake could be higher due to enhancement of the passive process, but this also affects peripheral uptake. For instance, neglecting cell-mediated uptake, $k_{t}=k_{b}=k_{p}=0$, and using $K_{bt}=yK_{bp}$, $K_{tb}=yK_{pb}$ with y=0.44, although possible to obtain similar tumor retention, results in fast NP clearance and very low uptake by peripheral organs (Figure 4c and inset). This contradicts several reported results that indicate a higher liver uptake (20–60%) with nanoparticles inside immune cells [15,26]. 

Furthermore, it is known that macrophage polarization in each organ is complex. Without specific knowledge of macrophage polarization states within organs, we assigned a single state to each compartment. This highlights a limitation of the current approach because instead of one polarization state (M1 or M2) in specific organs, one finds a distribution of M1/M2 cells in each organ, depending on the immune status and health of the host [27]. One might expect that the immune state of specific tissues influences the relative balance of (innate) immune cell populations to significantly affect nanoparticle uptake. In turn, the presence of nanoparticles in tissue will then likely affect immune composition in the tissue to induce concentration-dependent effects producing distinct changes in macrophage (and other immune cell) distributions, which can further influence nanoparticle retention. Such a time-dependent evolution of immune cell responses to the presence of nanoparticles is an interesting approach to consider in future modifications of PK/BD models. Note that different NP surface coatings can produce distinct NP uptake profiles (Figure 2), thus emphasizing the role of coating. 

The previously described simulations assumed the same macrophage state for peripheral and tumor (M1). In Figure 4d, we display results investigating the role of varying $K_{pm}$. A decrease of the nanoparticle uptake activity of macrophages in the periphery also generates a higher ITD. The number of NPs in peripheral organs decreases when $K_{pm}$ is lowered. For example, in this simulation the concentration of NPs in the periphery decreased to around 20%, lower than the 40% found for BNF-Plain, which resulted in a higher ITD for BNF-Her (value close to BNF-IgG), in agreement with experimental data [17]. 

Nevertheless, the effect of distinct macrophage polarization states on organs or tumor (due to environment or nanoparticle formulation) deserves more investigation, but indicates that more nanoparticle PK studies, especially those using non-invasive techniques, are urgently needed. A better understanding of NP uptake by phagocytic cells in tumors and organs will enhance clinical translation. Finally, our model is based on first-order kinetics. Recently, some have begun to investigate the role of nanoparticle liver uptake by Kupfer cells and its impact on BD by introducing enzyme-substrate-like modeling [26,28]. A similar approach can be adopted here to extend our proposed model. At present, the inclusion of non-linear effects is unsupported by the data.

Ultimately, the overall goal of the PK model proposed is to enable inferences of in vivo BD for a given NP. In particular, we focused our attention on NP uptake efficiency in tumor, using an in vivo PK profile and in vitro cell-specific internalization data using only macrophages and BNF^®^ nanoparticles. We selected macrophages as an initial system to model the effects of active biological uptake by tumor stromal compartments. A growing body of evidence demonstrates that both nanoparticle composition and multiple lineages of phagocytic immune (and stromal) cells interact in complex ways to affect the nanoparticle fate in vivo. Within the context of these limitations, and with the recognition that our initial approach represents a crude approximation of the whole, its use provides insights into the relative contributions on PK and tumor uptake of specific assumptions, which ultimately rely on data.

## 4. Materials and Methods

### 4.1. BNF Nanoparticles

Bionized nanoferrite^®^ (BNF^®^) nanoparticles were purchased from micromod Partikeltechnologie (Rostock, Germany). Unconjugated BNF^®^ nanoparticles (BNF-Plain) are hydroxyethyl starch-coated core-shell iron oxide nanoparticles that were used as received. Their synthesis and antibody conjugation has been described earlier [17]. Antibodies used for nanoparticle conjugation were trastuzumab, a humanized anti-HER2 monoclonal antibody (Her); and a polyclonal humanized nonspecific antibody (IgG). The physical characteristics of the nanoparticles used in this study are summarized in Table 5. 

### 4.2. Macrophage Culturing and Polarization 

The mouse macrophage cell line RAW264.7 was purchased from the American Type Culture Collection (ATCC, Manassas, VA, USA). RAW264.7 cells (passage number P3 to P5) were cultured in DMEM (Dulbecco’s Modified Eagle Medium), supplemented with high glucose (MilliporeSigma, Burlington, MA, USA) and 10% heat-inactivated serum (ThermoFisher Scientific Corporation, Waltham, MA, USA), with media changes every two days.

Murine IL-4 and IFN-γ for the induction medium (IM) was purchased from Miltenyi Biotec Inc. (San Diego, CA, USA). Lipopolysaccharide (LPS) was purchased from MilliporeSigma (Burlington, MA, USA). RAW264.7 cells of M0 (base) phenotype were induced to either M1 or M2 phenotypes by the addition of 100 ng/mL LPS and 50 ng/mL IFN-γ, or 10 ng/mL IL-4 to the media, respectively.

### 4.3. In Vitro Nanoparticle Uptake Experiments

RAW264.7 cells were pre-cultured in 10 cm Petri dishes for 1 h with normal media for the M0 phenotype and M1 or M2 induction media (IM) for the M1 or M2 phenotype polarization, respectively. The cells were then rinsed with PBS twice and harvested using a sterile cell scraper, collected in tubes, centrifuged at 1200 rpm for 5 min, and then counted (Cellometer Auto T4 Bright Field Cell Counter, Nexcelonm Biosciences, Lawrence, MA, USA). One million cells from each phenotype were suspended in 1 mL of the appropriate medium (normal or induction media) and divided into five treatment groups: BNF-Plain, BNF-HER, BNF-IgG, Herceptin plus BNF-Plain and IgG plus BNF-Plain. The macrophages were then treated with three different concentrations (0.125, 0.25 or 0.5 mg/mL Fe nanoparticles) for 3, 6, 12 or 24 h at 37 °C in an incubator (5% CO_2_). The Herceptin and IgG concentrations used were equivalent to those present on the conjugated BNF-HER and BNF-IgG. After incubation, tubes containing cells were centrifuged (1200 rpm for 6 min) to collect cells for a ferene-s assay. Measurements were performed in triplicate at 3 and 24 h, and in duplicate at 6 h. At 12 h, only one measurement was performed.

### 4.4. Intracellular Iron Quantification 

The total amount of iron taken up by the cells was determined by a colorimetric assay. The detailed protocol for conducting the modified ferene-s measurement of iron associated with cells after exposure to BNF nanoparticles has been previously described [29]. The stock reagent, working reagents, and standard reference materials were prepared beforehand. Ammonium acetate and glacial acetic acid were purchased from ThermoFisher Scientific Corporation (Waltham, MA, USA). The Fe standard reference material (SRM), FeCl_3_, Iron Standard for ICP (Inductively Coupled Plasma Spectroscopy), and 1000 ± 2 mg/L Fe in 2% nitric acid were purchased from Sigma-Aldrich (St. Louis, MO, USA), for calibrating the ferene-s assay. Briefly, 1 × 10^6^ macrophages were suspended in 1 mL media and were incubated at 37 °C during the different treatments with the occasional shaking or tapping of tubes to maximize distribution and prevent settling. After incubation, cells were centrifuged to separate them from free nanoparticles that remained in the supernatant, washed with PBS, and centrifuged a second time. This washing with PBS was repeated three additional times. The final cell pellet was resuspended in 1 mL of PBS and counted using a cellometer to estimate the total number of cells. A known number of cells in PBS was transferred into a 1 mL Eppendorf tube and centrifuged. The supernatant was discarded, and 1 mL of working solution was added to the cell pellet. Cell pellets were digested in the working solution by incubating them at room temperature for at least 20 h. Tubes were then centrifuged to separate solid cell debris, and the supernatant was analyzed with a UV/vis spectrophotometer at 595 nm. A calibration curve developed with known quantities of Fe using the SRM was used to estimate the iron concentration in the test samples.

### 4.5. Development of The Pharmacokinetic (PK) Model

PK model parameters are listed in Figure 5. $X_{0}$ is the injected dose (ID) of nanoparticles given in units of mass or %ID, with $X_{0}=100$ %ID. $K_{el}$ is the elimination or clearance rate, grouping all blood clearance mechanisms other than tumor uptake (given in units of time^−1^). The intercompartment exchange rate constants $K_{ij}$, are also given in units of time^−1^, with i and j being b for blood, p for peripheral, t for tumor, or *m* for macrophage. The constants $k_{i}$ (with i being b, p or t) are dimensionless multiplication factors. The quantities $x_{i}\left(t\right)$ and $m_{i}\left(t\right)$ are functions of time t, representing, respectively, the amount (also in units of mass, or in terms of %ID) of noninternalized nanoparticles in a given major compartment (dotted rectangles in Figure 5c), and the amount of nanoparticles internalized by macrophages in each of these major compartments (with macrophages assumed to be representative of all tumor-associated phagocytic immune cells). Thus, the total amount of nanoparticles within a given major compartment at time t corresponds to the sum $x_{i}\left(t\right)+m_{i}\left(t\right)$. 

#### 4.5.1. Estimating $K_{pb}$, $K_{el}$ and $K_{bp}$ from In Vivo PK and BD Data with the 2-Compartment Model

We hypothesized that modifying the standard 2-compartment model, accounting for active host biological processes with additional compartments, will improve model performance for predicting nanoparticle PK and BD. Thus, we began by estimating kinetics parameters for the host principle compartments (central and peripheral) from available in vivo data. We also assumed that nanoparticle PK and BD depends on the nanoparticle construct, thus we limited our search for in vivo data to BNF^®^ nanoparticles. 

Figure 5a is a schematic of a typical two-compartment PK model. It was used to derive the parameters $K_{pb}$, $K_{el}$ and $K_{bp}$. Mathematically, this can be represented by the following system of ODEs:(1)$$\frac{dx_{b}}{dt}=-\left(K_{el}+K_{bp}\right)x_{b}+K_{pb}x_{p}$$
(2)$$\frac{dx_{p}}{dt}=K_{bp}x_{b}-K_{pb}x_{p}$$
with the initial conditions being $x_{b}\left(0\right)=X_{0}$ and $x_{p}\left(0\right)=0$. The analytical solutions for $x_{b}\left(t\right)$ and $x_{p}\left(t\right)$ are:(3)$$x_{b}\left(t\right)=\frac{X_{0}}{\lambda_{b}-\lambda_{p}}\left[\left(\lambda_{b}-K_{pb}\right)e^{-\lambda_{b}t}-\left(\lambda_{p}-K_{pb}\right)e^{-\lambda_{p}t}\right]$$
(4)$$x_{p}\left(t\right)=\frac{K_{bp}X_{0}}{\left(\lambda_{p}-\lambda_{b}\right)}\left[e^{-\lambda_{b}t}-e^{-\lambda_{p}t}\right]$$
with $\lambda_{b}$ and $\lambda_{p}$ being rate constants such that
(5)$$\lambda_{b}+\lambda_{p}=K_{el}+K_{bp}+K_{pb}\;$$
(6)$$\lambda_{b}\lambda_{p}=K_{el}K_{pb}\;$$

To estimate the rate constants $\lambda_{b}$, $\lambda_{p}$ and $K_{pb}$ (we fitted Equation (3) to the normalized ($X_{0}=100$ %ID) blood PK data from in vivo experiments with BNF^®^ nanoparticles reported by Natarajan et al. [20]. We determined $K_{el}$ using Equation (6) from estimated values of $\lambda_{b}$, $\lambda_{p}$ and $K_{pb}$. Subsequently, $K_{bp}$ could be determined from Equation (5) to obtain values of $\lambda_{b}$, $\lambda_{p}$, $K_{pb}$, $K_{el}$ and $K_{bp}$ from fitting. 

#### 4.5.2. Estimating $K_{bt}$ and $K_{tb}$ from In Vivo BD Data 

The addition of the tumor compartment in Figure 5b implies the addition of two new rate constants to the model, namely: $K_{bt}$ and $K_{tb}$. The values for these rate constants were chosen to be such that the BD data reported by Natarajan et al. [20] matched simulations, especially for %ID retained within the tumor and peripheral tissues at 48 h for the 100-nm nanoparticles, i.e., about 1 and 40 %ID respectively. This occurs, for instance, when the values of $K_{bt}$ and $K_{tb}$ are chosen to be 0.05% of $K_{bp}$ and $K_{pb}$, respectively. 

#### 4.5.3. Estimating $K_{im}$ and $K_{mi}$ from In Vitro Data 

The PK model displayed schematically in Figure 5c is mathematically represented by the following system of first-order ordinary differential equations (ODEs):(7)$$\frac{dx_{b}}{dt}=-\left(K_{el}+K_{bp}+K_{bt}+k_{b}K_{bm}\right)x_{b}+K_{pb}x_{p}+K_{tb}x_{t}+k_{b}K_{mb}m_{b}$$
(8)$$\frac{dx_{p}}{dt}=K_{bp}x_{b}-\left(K_{pb}+k_{p}K_{pm}\right)x_{p}+k_{p}K_{mp}m_{p}$$
(9)$$\frac{dx_{t}}{dt}=K_{bt}x_{b}-\left(K_{tb}+k_{t}K_{tm}\right)x_{t}+k_{t}K_{mt}m_{t}$$
(10)$$\frac{dm_{b}}{dt}=k_{b}\left(K_{bm}x_{b}-K_{mb}m_{b}\right)$$
(11)$$\frac{dm_{p}}{dt}=k_{p}\left(K_{pm}x_{p}-K_{mp}m_{p}\right)$$
(12)$$\frac{dm_{t}}{dt}=k_{t}\left(K_{tm}x_{t}-K_{mt}m_{t}\right)$$
(13)$$\frac{dx_{b}}{dt}+\frac{dx_{p}}{dt}+\frac{dx_{t}}{dt}+\frac{dm_{b}}{dt}+\frac{dm_{p}}{dt}+\frac{dm_{t}}{dt}=-K_{el}x_{b}$$
with the initial conditions being $x_{b}\left(0\right)=X_{0}$, $x_{p}\left(0\right)=x_{t}\left(0\right)=m_{b}\left(0\right)=m_{p}\left(0\right)=m_{t}\left(0\right)=0$. Note that we assumed that the nanoparticles are eliminated exclusively via blood clearance. Indeed, we neglect biodegradation that nanoparticles could undergo after their internalization by macrophages. This assumption holds for the assessed timeframe, i.e., the first 48 h post-injection. 

The rate constants $K_{im}$ and $K_{mi}$ (Figure 5c, with i being either *b*, *p*, or *t*) were determined from in vitro internalization experiments with RAW264.7 macrophages described. $K_{in}$ and $K_{out}$ are the rates at which nanoparticles enter or leave the macrophage intracellular medium, respectively. $x_{out}\left(t\right)$ represents the amount of noninternalized nanoparticles, while $x_{in}\left(t\right)$ represents the amount of internalized nanoparticles. This can be mathematically represented by the following system of differential equations:(14)$$\frac{dx_{out}}{dt}=-K_{in}x_{out}+K_{out}x_{in}\;$$
(15)$$\frac{dx_{in}}{dt}=K_{in}x_{out}-K_{out}x_{in}\;$$
with the initial conditions being $x_{out}\left(0\right)=X_{0}$ and $x_{in}\left(0\right)=0$, where $X_{0}$ represents the amount of nanoparticles initially introduced in the cell culture. The analytical solutions for $x_{out}\left(t\right)$ and $x_{in}\left(t\right)$ are:(16)$$x_{out}\left(t\right)=\frac{X_{0}}{\left(K_{in}+K_{out}\right)}\left[K_{out}+K_{in}e^{-\left(K_{in}+K_{out}\right)t}\right]\;$$
(17)$$x_{in}\left(t\right)=\frac{K_{in}X_{0}}{\left(K_{in}+K_{out}\right)}\left[1-e^{-\left(K_{in}+K_{out}\right)t}\right]\;$$

We note that for the timeframe of our studies (up to 24 h in vitro, and up to 48 h in vivo), any biodegradation of internalized nanoparticles was neglected. Mathematically, this is evidenced by the steady state solutions (t→∞):(18)$$x_{out}\left(t\to \infty \right)=\frac{K_{out}X_{0}}{\left(K_{in}+K_{out}\right)}\;$$
(19)$$x_{in}\left(t\to \infty \right)=\frac{K_{in}X_{0}}{\left(K_{in}+K_{out}\right)}\;$$
which are nonzero constants in time (rather than zero if biodegradation was considered). 

### 4.6. Simulations, Fitting and Statistical Analysis of Data 

All numeric simulations were run in Maple 13 (Maplesoft, Waterloo, Ontario, CA). The fitting of data to model equations was performed using a weighted least square procedure with OriginPro 2015 software (OriginLab Corp., Northampton, MA, USA). The numerical fitting method minimizes the sum of the square errors and calculates values and variance by assigning a statistically meaningful weighting for data. The fitting of other data to relevant model parameters was performed as described.

## 5. Conclusions

In this study, we proposed a six-compartment PK model to simulate a tumor-associated immune cell-mediated retention mechanism for systemically delivered nanoparticles. In this preliminary attempt, we assumed that macrophages are solely responsible for the proposed process. Published in vivo PK and BD data were combined with data obtained from in vitro cell-internalization experiments with macrophages to guide simulations of BNF-Plain and BNF-Her NP PK to gain insights into differences between in vitro and in vivo settings. The validity of the macrophage-approximation bears further testing with other nanoparticulate systems, and data using other lineages of phagocytic cells will provide additional insights to refine the models.

## Figures and Tables

**Figure 1 ijms-23-15664-f001:**
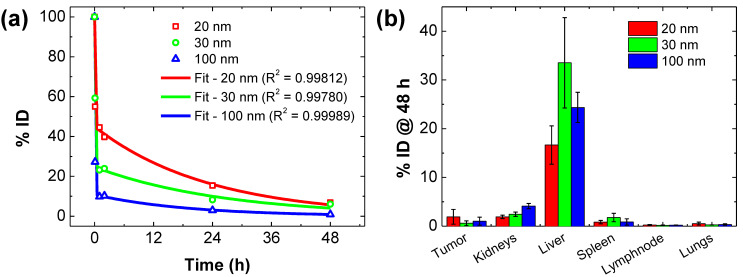
In vivo pharmacokinetic (PK) and ex vivo biodistribution (BD) for bionized nanoferrite^®^ (BNF^®^) nanoparticles. Except for fitting results, figures are reproduced from data reported by Natarajan et al. [20]. (**a**) Fitting (solid lines) to data using the analytical expression for $x_{b}\left(t\right)$ (Equation (10)) in text, Materials and Methods) to the in vivo blood PK data obtained from BNF^®^ nanoparticles having average diameters of 20, 30 and 100 nm. Experimental conditions for blood collection are described in the cited reference. (**b**) Ex vivo biodistribution profiles of BNF^®^ nanoparticles 48 h after injection, as reported in the cited reference. BD data reported as % injected dose (%ID)/g were converted to %ID. Adapted with permission from data previously reported by Natarajan, et al. [20]. Copyright 2008 American Chemical Society.

**Figure 2 ijms-23-15664-f002:**
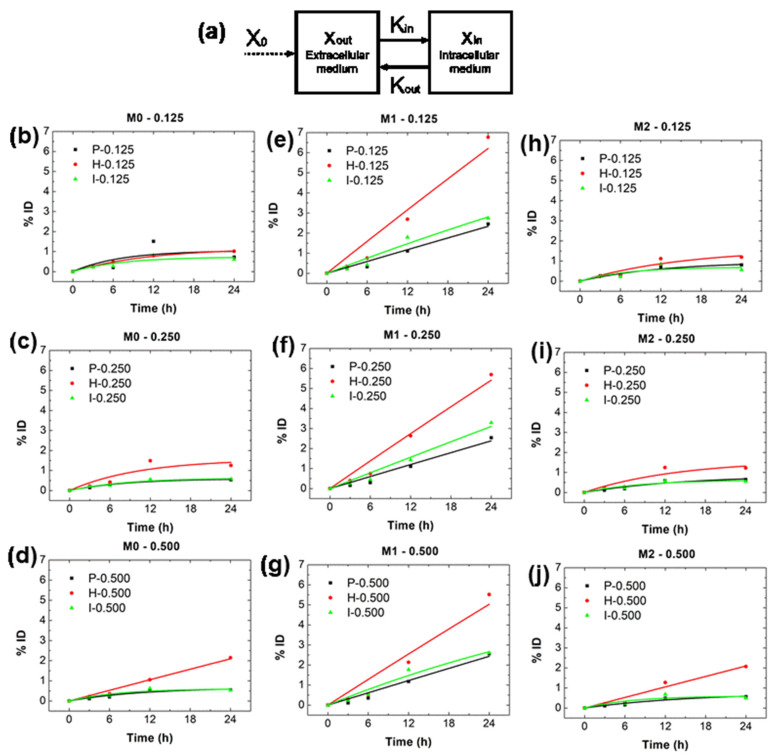
Two-compartment model results using data obtained from in vitro cell-internalization experiments with macrophages. (**a**) Two-compartment model used to determine the rate constants $K_{in}$ and $K_{out}$, i.e., to model the macrophage internalization kinetics in vitro. (**b**–**j**) Fitting to mean values obtained from in vitro cell-internalization experiments with macrophages. Fitting of the macrophage internalization kinetics $x_{\mathrm{in}}\left(t\right)$ was conducted by weighted least squares fitting using Equation (17) to the in vitro mean experimental values (ferene-s assay at 3, 6, 12, and 24 h) described above. RAW264.7 murine macrophage cells were incubated with one of three different BNF initial concentrations (0.125, 0.250, or 0.500 mg of Fe) and three different BNF configurations (P = BNF-Plain, black lines, no antibodies; H = BNF-Her, red lines; and I = BNF-IgG, green lines) as described. (**b**–**d**) Results for nonpolarized macrophages, M0. (**e**–**g**) Results for M1 macrophages. (**h**–**j**) And, results for M2 macrophages (values obtained using complete data set are provided in Appendix A).

**Figure 3 ijms-23-15664-f003:**
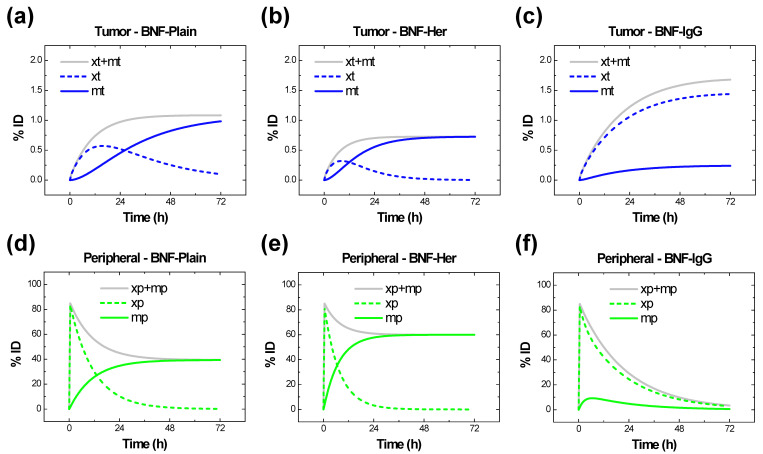
Simulated PK profiles for 100-nm BNF^®^ nanoparticles. Simulated tumor and peripheral retentions of 100-nm BNF-Plain (**a**,**d**), BNF-Her (**b**,**e**), and BNF-IgG (**c**,**f**) nanoparticles. Dashed lines represent passive retention, solid-colored lines represent active retention by macrophages, and solid-gray lines represent the sum of passive and active retentions. Simulations suggest that resident macrophages dominate tumor and peripheral active retention of BNF-Plain and BNF-Her nanoparticles about 12 h after administration. Meanwhile, passive retention of BNF-IgG would be more significant.

**Figure 4 ijms-23-15664-f004:**
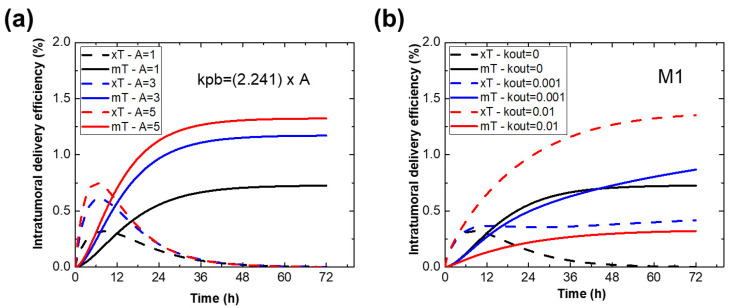
Simulated PK profiles for BNF-Her nanoparticles. Simulated tumor retentions of 100-nm BNF-Her nanoparticles varying several model parameters (**a**) $K_{pb}$, (**b**) $K_{out}$ from macrophages M1, and (**d**) $K_{pm}$. Dashed lines represent passive retention ($x_{i}$), and solid-colored lines represent active retention by macrophages ($m_{i}$). (**c**) Black, red and blue lines represent, respectively, the kinetics of NPs in the blood, peripheral and tumor compartment, in %ID, for the case without cell retention, i.e., $k_{t}=k_{b}=k_{p}=0$ and *y* = 0.44. The inset shows intratumor delivery efficiency.

**Figure 5 ijms-23-15664-f005:**
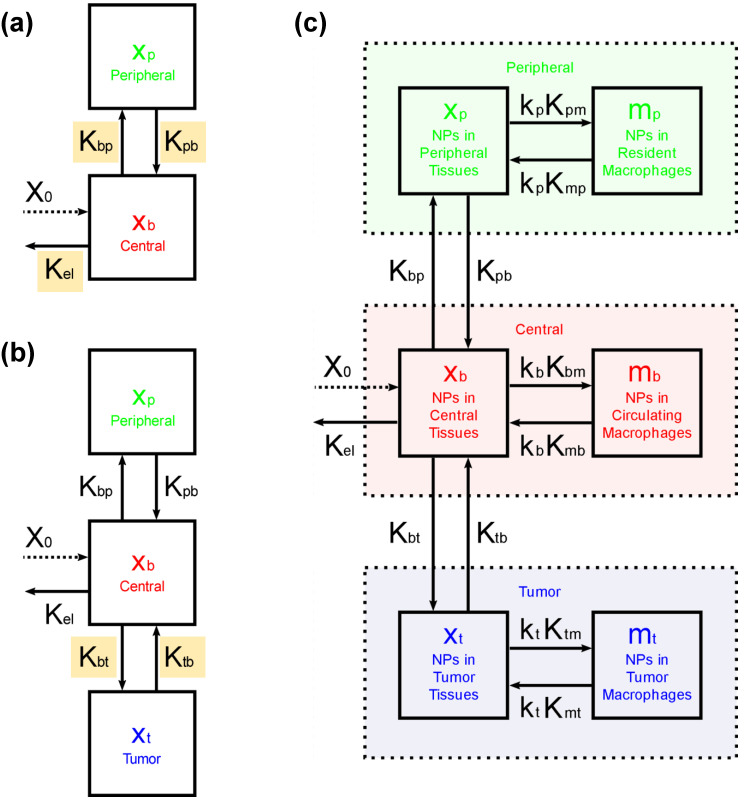
Pharmacokinetic (PK) compartment models. (**a**) Typical two-compartment model used to describe the PK of a small molecule drug. (**b**) Three-compartment model used within the context of cancer therapies to differentiate the tumor specific uptake from the nonspecific uptake by healthy peripheral tissues and organs, where $x_{i}$ (with i=b for blood, p, for peripheral, and t for tumor) represents the amount of nanoparticles either in the central, peripheral or tumor compartment, respectively. The intercompartment exchange rate constants $K_{ij}$ are given in units of time^−1^. $X_{0}$ represents the injected dose, and $K_{el}$ denotes the rate at which nanoparticles are eliminated from the organism. (**c**) Proposed six-compartment model intended to quantitatively model the role of tumor-associated immune cells on tumor uptake. Here $x_{i}$ represents the amount of noninternalized nanoparticles within one of the three major compartments (dotted rectangles), whereas $m_{i}$ represents the amount of nanoparticles internalized by macrophages in each of the major compartments (with macrophages here being representative of all tumor-associated phagocytic immune cells as a first approximation). $k_{i}$ are dimensionless multiplication factors used to match the in vitro kinetics with the observed in vivo results.

**Table 1 ijms-23-15664-t001:** Values of $K_{pb}$, $K_{el}$ and $K_{bp}$ for bionized nanoferrite^®^ (BNF^®^) nanoparticles. Adapted with permission from Natarajan et al. [20]. Copyright 2008 American Chemical Society.

Diameter [nm]	$\lambda_{b}\;[h^{-1}]$	$\lambda_{p}\;[h^{-1}]$	$K_{pb}\;[h^{-1}]$	$K_{el}\;[h^{-1}]$	$K_{bp}\;[h^{-1}]$
20	20.169 ± 1.999	0.043 ± 0.003	9.085 ± 1.039	0.095 ± 0.016	11.032 ± 2.253
30	09.304 ± 0.641	0.038 ± 0.006	2.325 ± 0.238	0.151 ± 0.031	06.866 ± 0.685
100	20.273 ± 0.348	0.052 ± 0.005	2.241 ± 0.094	0.471 ± 0.050	17.613 ± 0.364

**Table 2 ijms-23-15664-t002:** Values of $K_{in}$ and $K_{out}$ obtained from the fitting shown in Figure 2 for BNF-plain using mean values.

BNF-Plain					
		0.125	0.250	0.500	Average *
**M0**	$K_{in}$	0.00146 ± 0.00150	0.00070 ± 0.00012	0.00061 ± 0.00022	0.00092 ± 0.00051
	$K_{out}$	0.13862 ± 0.20191	0.11676 ± 0.03122	0.09149 ± 0.05568	0.11562 ± 0.07059
**M1**	$K_{in}$	0.00099 ± 0.00023	0.00101 ± 0.00029	0.00103 ± 0.00027	0.00101 ± 0.00015
	$K_{out}$	0.00000 ± 0.02115	0.00000 ± 0.02621	0.00000 ± 0.02408	0.00000 ± 0.01380
**M2**	$K_{in}$	0.00079 ± 0.00021	0.00057 ± 0.00018	0.00048 ± 0.00014	0.00061 ± 0.00010
	$K_{out}$	0.08162 ± 0.03797	0.06572 ± 0.04289	0.06319 ± 0.03955	0.07018 ± 0.02320

* Average of the values found for each initial dose (0.125, 0.250, and 0.500 mg of Fe).

**Table 3 ijms-23-15664-t003:** Values of $K_{in}$ and $K_{out}$ obtained from the fitting shown in Figure 2 for BNF-Her using mean values.

BNF-Her					
		0.125	0.250	0.500	Average *
**M0**	$K_{in}$	0.00103 ± 0.00007	0.00150 ± 0.00075	0.00088 ± 0.00014	0.00114 ± 0.00025
	$K_{out}$	0.08675 ± 0.01000	0.09434 ± 0.07842	0.00000 ± 0.01436	0.06036 ± 0.02678
**M1**	$K_{in}$	0.00267 ± 0.00091	0.00232 ± 0.00058	0.00215 ± 0.00080	0.00238 ± 0.00045
	$K_{out}$	0.00000 ± 0.03106	0.00000 ± 0.02281	0.00000 ± 0.03412	0.00000 ± 0.01716
**M2**	$K_{in}$	0.00103 ± 0.00040	0.00114 ± 0.00052	0.00088 ± 0.00026	0.00102 ± 0.00024
	$K_{out}$	0.06352 ± 0.05048	0.06970 ± 0.06238	0.00000 ± 0.02754	0.04441 ± 0.02828

* Average of the values found for each initial dose (0.125, 0.250, and 0.500 mg of Fe).

**Table 4 ijms-23-15664-t004:** Values of $K_{in}$ and $K_{out}$ obtained from the fitting shown in Figure 2 for BNF-IgG using mean values.

BNF-IgG					
		0.125	0.250	0.500	Average *
**M0**	$K_{in}$	0.00100 ± 0.00050	0.00075 ± 0.00018	0.00083 ± 0.00029	0.00086 ± 0.00020
	$K_{out}$	0.13548 ± 0.09770	0.11884 ± 0.04218	0.13587 ± 0.06790	0.13006 ± 0.04208
**M1**	$K_{in}$	0.00129 ± 0.00038	0.00131 ± 0.00033	0.00137 ± 0.00036	0.00132 ± 0.00021
	$K_{out}$	0.00682 ± 0.02855	0.00000 ± 0.02288	0.01692 ± 0.02647	0.00791 ± 0.01505
**M2**	$K_{in}$	0.00102 ± 0.00064	0.00074 ± 0.00024	0.00086 ± 0.00045	0.00087 ± 0.00027
	$K_{out}$	0.14503 ± 0.12777	0.11689 ± 0.05677	0.14133 ± 0.10473	0.13442 ± 0.05823

* Average of the values found for each initial dose (0.125, 0.250, and 0.500 mg of Fe).

**Table 5 ijms-23-15664-t005:** Physical characteristics of the nanoparticles.

Nanoparticle	Lot Number	Mean Hydrodynamic Diameter [nm]	Polydispersity Index (PDI)	Zeta Potential [mV]	Protein Concentration [µg/mg]
BNF-Plain	0901810	102.3	0.115	−3.4 ± 7.0	N/A
BNF-HER	1261810	140.9	0.117	−6.6 ± 3.6	32.6
BNF-IgG	0981810	130.0	0.109	−5.1 ± 6.6	35.5

## Data Availability

Not applicable.

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
