# Peer review of "A Predictive Pharmacokinetic Model for Immune Cell-Mediated Uptake and Retention of Nanoparticles in Tumors"

_ijms, 2022, doi:10.3390/ijms232415664_

Round 1

Reviewer 1 Report

The manuscript under evaluation reports the development of a pharmacokinetic model to simulate the tumor-associated immune cells retention of systemically delivered nanoparticles. Even though the study is presenting some interesting results, it should be revised according to the following comments:

Comment 1:

The content presentation should be improved. The subsections must be well organized, clearly presented and numerated.

Comment 2:

For the conducted in vitro test, the authors do not mention the replicates or the reproducibility of the experiments. No comparative statistical analysis has been done neither a normality test to determine whether sample data has been drawn from a normally distributed population.

Comment 3:

The authors should obviously mention and discuss the main limitations of the proposed approach and give their suggestions for its future optimization. This would be of great interest for the readers.

Author Response

Response to reviewer comments:

We, the authors thank the reviewers for taking time to read our manuscript and to provide suggestions for its improvement. Below, we provide point-by-point responses to reviewer comments. Changes to manuscript text are in red font.

Reviewer 1:

Comment 1: The content presentation should be improved. The subsections must be well organized, clearly presented and numerated.

Response: We thank the reviewer for the helpful comment and we have addressed this by reorganizing the Methods and Results sections to remove text describing methods from the latter section, enumerating the subsections, and by consolidating and revising subsection titles to better communicate the subject to readers.

Comment 2: For the conducted in vitro test, the authors do not mention the replicates or the reproducibility of the experiments. No comparative statistical analysis has been done neither a normality test to determine whether sample data has been drawn from a normally distributed population.

Response: We thank the reviewer for pointing out our oversight. We have revised the Methods section in the text to inform readers of numbers of replicate in vitro experiments. We have also revised the Statistical Analysis to describe how data were analyzed. Our results are provided in text and supplementary materials.

Comment 3: The authors should obviously mention and discuss the main limitations of the proposed approach and give their suggestions for its future optimization. This would be of great interest for the readers.

Response: This is a most important topic, particularly in such exploratory works. We have expanded discussion of this topic in the Discussion section. 

Reviewer 2 Report

I'm invited to review the manuscript "A Predictive Pharmacokinetic Model for Immune Cell-Mediated Uptake and Retention of Nanoparticles in Tumors.

I found following major issues in the paper:

1. All the authors of this manuscript have self-cited their papers more than 10 times which is suspicious. Provide the strong reason for this. 

2. in supplementary file there are tow images and one word file. Same images are present even in word file. 

3. In introduction: line 69, 74 and 116 there are citation with images number and image is also provided in introduction part. Justify this. Instead add this in methodology section. 

4. Paragraph 122-134 is completely from single reference [17]. which is one of your self-citation. Justify its inclusion 

Minor comment

line 57-58: for this single sentence 5 references are cited. rewrite this sentence and cite 2-3 references 

Author Response

Response to reviewer comments:

We, the authors thank the reviewers for taking time to read our manuscript and to provide suggestions for its improvement. Below, we provide point-by-point responses to reviewer comments. Changes to manuscript text are in red font.

Reviewer 2:

Comment 1: All the authors of this manuscript have self-cited their papers more than 10 times which is suspicious. Provide the strong reason for this. 

Response: We thank the reviewer for pointing out our oversight. We have revised the text to reduce the number of citations of our previously published works. That said, we point out that both the motivation for the present study, and in vivo PK and BD data using the BNF nanoparticles is from previously published works by us. We have, as much as possible cited the most recent works by others showing that active biological processes (e.g. macrophages, endothelial cells, etc.) within the tumor microenvironment engage to determine nanoparticle fate. However, with regard to the PK and BD of BNF nanoparticles used in the current study, there is no other publication(s) to our knowledge. We have accordingly revised the text in the Discussion section to highlight this limitation and its implications (see also response to Reviewer 1).

Comment 2: In supplementary file there are tow images and one word file. Same images are present even in word file.

Response: We thank the reviewer for pointing out our oversight. The supplementary material has new figures, Fig. S1, S2, S3 and S4.  We also included Tables S1 and S2. Table S1 correspond to the mean analysis, while Table S2 is the all data analysis. See response to reviewer 1.

Comment 3: In introduction: line 69, 74 and 116 there are citation with images number and image is also provided in introduction part. Justify this. Instead add this in methodology section. 

Response: We thank the reviewer for pointing out our oversight. We have corrected this. We have removed references to figures in the Introduction, and modified the Methods to focus reader attention on the figure (Figure 1) in the relevant portion of this section.

Comment 4: Paragraph 122-134 is completely from single reference [17]. which is one of your self-citation. Justify its inclusion 

Response: We have deleted this paragraph.

Comment 5: Minor comment - line 57-58: for this single sentence 5 references are cited. rewrite this sentence and cite 2-3 references 

Response: We have revised this in response to reviewer comments.

Round 2

Reviewer 1 Report

The authors have satisfactorily addressed my concerns mentioned in the first review report.

Reviewer 2 Report

All the comments are addressed satisfactorily.